# Tobacco Smoke Exposure in Pregnant Dogs: Maternal and Newborn Cotinine Levels: A Pilot Study

**DOI:** 10.3390/vetsci10050321

**Published:** 2023-04-28

**Authors:** Giulia Pizzi, Silvia Michela Mazzola, Alessandro Pecile, Valerio Bronzo, Debora Groppetti

**Affiliations:** Department of Veterinary Medicine and Animal Sciences, Università degli Studi di Milano, 26900 Lodi, Italy; giulia.pizzi@unimi.it (G.P.); silvia.mazzola@unimi.it (S.M.M.); alessandro.pecile@unimi.it (A.P.); valerio.bronzo@unimi.it (V.B.)

**Keywords:** canine, amniotic fluid, hair, nicotine, birth

## Abstract

**Simple Summary:**

During the last decades, the detrimental effects of exposure to tobacco smoke on children and mothers have been emphasized. Conversely, not as much attention has been paid to the risks for pets, even though second-hand smoke can affect them too. Cotinine, the primary metabolite of nicotine, has recently been found in the serum and hair of dogs living with smoking owners. The purpose of this study was to investigate the possible transplacental crossing of cotinine in bitches exposed to passive smoke during pregnancy by measuring cotinine in amniotic fluid, newborn hair, and maternal hair and serum at birth. Cotinine was measurable in all matrices and samples collected during the Caesarean section, with higher concentrations in exposed dogs than those not exposed. These findings highlight the transfer of cotinine from mother to fetus and warn that owners should be more aware of possible adverse health consequences for puppies.

**Abstract:**

Active and passive smoking in pregnant women is associated with perinatal morbidity and mortality risk, including abortion, preterm birth, low birthweight, and malformations. No data are available on intrauterine exposure to smoking during pregnancy in dogs. This study aimed to fill this gap by exploring the detectability and quantity of cotinine, the major metabolite of nicotine, in maternal (serum and hair) and newborn (amniotic fluid and hair) biospecimens collected at birth in dogs. For this purpose, twelve pregnant bitches, six exposed to the owner’s smoke and six unexposed, were enrolled. A further six non-pregnant bitches exposed to passive smoke were included to investigate the role of pregnancy status on cotinine uptake. Exposed dogs, dams, and puppies had greater cotinine concentrations than unexposed ones. Although without statistical significance, serum and hair cotinine concentrations were higher in pregnant compared to non-pregnant bitches, suggesting a different sensitivity to tobacco smoke exposure during gestation. The present results provide evidence for cotinine transplacental passage in the dog. It is conceivable that fragile patients such as pregnant, lactating, and neonate dogs may be more susceptible to the harmful effects of second-hand smoke exposure. Owners should be sensitized to the risk of smoke exposure for their pets.

## 1. Introduction

Cigarette smoking is a significant public health issue worldwide. A large number of published studies have addressed the toxic effects of active smoking. On the contrary, the impact of passive smoking is not fully understood. However, both active and passive smoking are held equally responsible for an increased risk of various types of cancer, as well as nervous, endocrine, immune, respiratory, and cardiovascular disease, allergic rhinitis and dermatitis, as well as food sensitivities [1]. There is no safe exposure to tobacco smoke. Second-hand smoke (SHS, the side stream smoke coming from the burning end of the cigarette combined with that exhaled by the mainstream smoke that smokers exhale) and third-hand smoke (THS, the chemical contaminant particles emitted into the air after tobacco smoking, which settle on and adhere to surfaces from where they can float and then be released back into the air, undergo chemical change transformations, or accumulate and be embedded in materials) both contain many hazardous substances. Therefore, individuals involuntarily exposed to SHS and THS are also adversely affected [2]. In humans, the noxious influence of maternal smoking on fetal and neonatal health is well recognized and includes abortion, perinatal death, preterm birth, intrauterine growth retardation, behavioral and cognitive issues, and cryptorchidism [3,4,5,6,7,8]. In fact, nicotine causes vasoconstriction that, in pregnancy, also involves the umbilical cord, leading to a reduced oxygen supply and impairment of placental development and function [9]. While smoking is a conscious and dismissible action, exposure to environmental tobacco smoke is more insidious and often inevitable, particularly in a domestic setting. Like active smokers, even pregnant women exposed to second-hand and third-hand smoke have a greater risk of preterm birth [10] and low birth weight babies [11]. Additionally, congenital malformations such as orofacial clefts and neural tube defects occurred more frequently in babies whose mothers experienced passive smoking during pregnancy [12] due to nicotine-induced damage to fetal DNA [13]. Furthermore, environmental exposures such as tobacco smoke during early life, particularly the in utero period, can influence health and vulnerability to disease in adulthood [14].

Cigarette smoking produces toxic compounds, including carbon monoxide, formaldehyde, nicotine, and many carcinogens. When tobacco smoke constituents reach the lung’s alveoli, they are rapidly absorbed and transferred through the bloodstream and extensively distributed into body tissues [15]. Tobacco addiction is caused by nicotine, the most specific component of cigarette smoke (1–2 mg per cigarette). Nicotine is metabolized into more than 20 different derivatives, but in humans, 70% of nicotine is oxidized to cotinine by the liver [16]. Nicotine and its metabolites were described as effective markers of both direct and indirect smoke exposure. However, the fluctuation in blood levels and the longer persistence in the body (two hours half-life) make nicotine poorly suited for monitoring chronic smoke exposure, preferring instead the measurement of cotinine [17]. Therefore, cotinine is considered the analyte of choice to detect tobacco exposure since it fulfils the prerequisites of specificity and retention time (18–20 h). Moreover, it has been found in humans at detectable levels in the blood, hair, urine, saliva, nasal fluid, amniotic fluid, breast milk, and neonatal hair [18,19,20,21,22,23]. It can help to quantify tobacco exposure in actively and passively exposed individuals and is considered superior to thiocyanate as a biomarker in validating cigarette smoking [23]. Since it pools fetal excretions and maternal secretions, amniotic fluid appeared to be an excellent tool for detecting intrauterine exposure. At the same time, neonatal hair proved to be an effective non-invasive biospecimen to estimate smoke exposure over the last trimester of pregnancy [21].

Unlike humans, the effect of environmental smoke exposure on canine pregnancy is poorly investigated, and information on dams and puppies is not available. Since they closely share their lifestyle with their owners, domestic animals may be regarded as potential victims of second- and third-hand smoke. However, owners and breeders do not seem to perceive or be aware of the risk of smoke exposure to their pets. The present study aimed to detect and quantify cotinine in maternal and neonatal biospecimens. In particular, cotinine concentration in maternal serum and hair, along with neonatal hair and amniotic fluid collected at birth, was measured and compared between bitches exposed and non-exposed to cigarette smoke. Lastly, the hypothesis of different susceptibility to smoke exposure due to pregnancy status was explored by comparing cotinine concentrations in the serum and hair of pregnant and non-pregnant bitches exposed to the owner’s smoke.

## 2. Materials and Methods

This study was approved by the Ethical Committee of the Università degli Studi di Milano (OPBA_77_2017) and conducted with the consent of the dog owners. Amniotic fluid was collected at birth during Caesarean-section on residual fluid waste in the canine species.

### 2.1. Animals

Eighteen bitches referred to the Reproduction Unit of the Veterinary Teaching Hospital of the Università degli Studi di Milano were enrolled. Only householding bitches undergoing C-sections were included. C-sections were performed due to breed predisposition to dystocia, previous history of uterine inertia, litter size, and/or maternal age.

Based on passive smoke exposure, animals were divided into three groups: exposed pregnant bitches (n = 6, EXp), not exposed pregnant bitches (n = 6, NEp), and exposed non-pregnant bitches (n = 6, EXnp). Patients were considered subjected to tobacco smoke if the owner’s indoor consumption was smoking at least one cigarette in the house every day per day in the last two months. The intensity of passive smoke was semi-quantitatively scored as follows: “+” up to five cigarettes per every day; “++” more than five cigarettes per day. Dogs were defined as not exposed if they lived with non-smoking owners.

Breed, age, and body weight were noted as well as litter size in the case of pregnant dogs. In the EXp and NEp groups, bitches were monitored from proestrus to parturition. At birth, puppies underwent routine neonatal care with emergency procedures provided in case of critical conditions. Mortality at birth and any malformations were recorded. Group EXnp bitches were recruited among those belonging to the Veterinary Teaching Hospital for routine clinical procedures.

### 2.2. Samples Collection

Maternal blood and hair samples were collected immediately before anaesthesia induction for C-sections in parturient dogs. In the EXnp group, samples were taken from the remaining material after screening analysis. The forelimb hair was shaved close to the skin with a clipper before placing the IV catheter and stored into a paper envelope at r/t until analysis. Blood sample collection (1.5 mL) was performed from the cephalic vein into serum tubes (Vacutest, Securlab, Roma, Italy), immediately centrifuged (5 min at 3500× *g*), and stored at −20 °C for further analysis.

At the time of the puppy extraction, amniotic fluid was collected from each puppy using a 20 mL sterile syringe, then pooled for each litter and stored at −20 °C until analysis.

The puppies’ hair was clipped from the ventral surface of the tail base. The littermates’ hair was pooled together and stored in a paper envelope at r/t until analysis.

### 2.3. Cotinine Assay

Cotinine extraction from maternal and neonatal hair was performed following the procedure reported in the literature by Gunay et al. (2010) [24] and recently described in dogs by our group [25]. Hair, serum, and amniotic cotinine samples were analysed following the manufacturer’s instructions, using an enzyme-linked immunosorbent assay kit (CET058Ge, Cloud Clone, Katy, TX 77494, USA). The commercially available kit was designed to quantitatively measure the cotinine levels in various sample biospecimens. According to pre-experiment pre-test outcomes and results, and the assay specification as recommended by the manufacturer’s recommendations, serum and amniotic samples were not diluted. Samples analysis was performed in duplicate, and absorbance was measured in a microplate reader using a wavelength of 450 nm (Multiskan EX, LabSystem, Thermo Fisher Scientific, Milan, Italy). Regression analysis was used to calculate the concentration based on the relevant standard curves (range of 50 pg/mL to 617.3 pg/mL). The mean recovery was 98% ± 7.0. The laboratory researcher was blinded to not be aware of the hypotheses’ assumptions and conditions.

### 2.4. Statistical Analysis

Data analysis was performed using a statistical program (IBM SPSS 28.0, Armonk, NY, USA). Descriptive statistics for quantitative variables were expressed as a range, mean, and standard deviation. Since data of continuous variables were not normally distributed (Shapiro-Wilk test), the results were compared with a non-parametric post-hoc pairwise median test for independent samples. Statistical significance was set at *p* < 0.05 and levels of *p* < 0.001 were regarded as highly significant.

## 3. Results

### 3.1. Clinical Outcomes

The breed, age, body weight, and litter size of our caseload are summarized in Table 1.

Basset hounds (33%) and Bernese Mountain Dogs (22.2%) were the most represented breeds. Bitches aged from 1.5 to 12 years (34.3 ± 3.1) and weighed 14 to 63.6 kg (31.5 ± 13.5). Litter size ranged from 2 to 8 pups (5.1 ± 2.4). A total of 61 puppies were delivered, 33 from dams exposed to passive tobacco smoke and 28 from not exposed ones. Neonatal mortality in the EXp and NEp groups was the same, i.e., two stillborn puppies and two further puppies that died within 30 days of life. Similarly, one puppy in the EXp group showed a malformation (renal dysplasia), and one puppy in the NEp group suffered from a cleft lip and palate.

### 3.2. Cotinine Detection

Cotinine was detectable in all serum, hair, and amniotic samples of both exposed and non-exposed dogs, in dams and puppies. Calibration curves for each matrix were developed following the manufacturer’s instructions.

Considering the whole caseload, cotinine concentration ranged from 0.87 to 24.6 ng/mL in serum, 1.3 to 40.8 ng/mg in bitches’ hair, 0.4 to 14.9 ng/mL in amniotic fluid, and 1.27 to 24 ng/mg in newborn hair. The cotinine concentration in maternal serum was positively correlated with that in maternal and newborn hair and in amniotic fluid (*p* < 0.0001; Figure 1a,b).

Pregnant bitches exposed to cigarette smoke (EXp) and their offspring had cotinine concentrations significantly greater than pregnant dogs not exposed (NEp) as well as their neonates in all the biospecimens (*p* = 0.004; Figure 2).

Serum and hair cotinine concentrations were higher in pregnant than non-pregnant bitches exposed to passive tobacco smoke but without a statistical significance (Figure 3).

The intensity of smoke exposure did not influence all the matrices and the concentration of cotinine was not affected by the number of cigarettes smoked.

Cotinine concentration was not correlated with maternal age, body weight, and litter size. Moreover, no differences in maternal age, body weight, and litter size were evidenced between exposed and non-exposed groups.

## 4. Discussion

Starting from ‘Barker’s hypothesis’, the theory of fetal programming, which underlines the role of the in uterus environment on the development of disease in adult life, is becoming increasingly topical in both animals and humans. Events occurring during the embryonic and fetal period, including exposure to smoke, may have significant effects on lifelong health, leading to obesity, type II diabetes, cardiovascular disease, hypertension, and tumors [14]. Pregnancy is a special moment during which mother and fetus share the same space, stressors, and environmental factors, together with nutrients and oxygen via blood and placental exchanges [26,27]. This mutualism also concerns tobacco smoke exposure through which nicotine, the principal cigarette component, can enter the maternal body by inhalation, skin absorption, and ingestion of residual particles settled on surfaces and hair [28]. Almost 80% of the absorbed nicotine is converted to cotinine [29], which can easily cross the human placental barrier and reach the fetus [30]. Acting as a collector of maternal and fetal secretions, the amniotic fluid contains cotinine whose concentrations are proportional to the degree of a woman’s smoke exposure [31]. Despite the several publications addressed to humans [2,32,33], to our knowledge, this study is the first to measure canine cotinine concentration in dams and puppies exposed to passive smoke during pregnancy.

Our results demonstrate that cotinine is detectable in the canine amniotic fluid and hair of newborn puppies and in maternal hair and serum. These findings confirmed the transplacental transfer of cotinine due to intrauterine exposure in passive smoker pregnant bitches. The presence of cotinine in the hair collected immediately after birth from the puppies showed its early fetal accumulation during pregnancy. Hair cotinine concentration was about seven-fold higher in puppies whose dams were exposed to tobacco smoke than in those who were not. Similarly, in humans, babies with both active and passive smoking mothers incorporate cotinine derived from fetal blood into hair fibers [34].

Amniotic cotinine was significantly higher in bitches exposed to cigarette smoke compared to not exposed ones but with a lower concentration than in maternal serum. On the contrary, in women, amniotic cotinine prevailed over that in maternal serum, proving an accumulation in fetal fluid [22,31]. Similar divergences emerged when comparing pollutant exposure in human and canine placenta, with dogs showing a lower bioaccumulation capacity and a higher excretion rate [35]. The different placentation between humans and dogs can explain this discrepancy. Despite companion animals being regarded as valuable models for humans because they share the same environment, stressors, disease predisposition, and many genetic traits, species-specific anatomo-morphological and metabolic differences may affect the molecules’ passage through the placenta [35,36]. The hemochorial placenta of the woman is characterized by a limited cellular barrier and an intimate relationship between maternal vessels and the trophoblast [37] while the canine endotheliochorial placenta has a greater number of tissue layers separating maternal from fetal circulation [38]. Trans-placental transfer depends not only on the thickness of the cell barrier between the maternal and fetal circulation but also on the structure of the maternal-fetal interface [35]. For example, fatty acids and keto acids readily cross the hemochorial but not the epitheliochorial placenta [35]. In addition, the passage of iron occurs by penetration in the hemochorial placenta, by phagocytosis in the endotheliochorial placenta, and by secretion in the epitheliochorial placenta [35]. Furthermore, protein transporters can be differently localized, regulated, and responsive to inhibitors, and move dissimilar substrates. In addition, fetal vascularization, pregnancy length, and the endocrine and transporting functional zone of the placenta may differ among species [36]. These anatomical and functional aspects together with our results seem to suggest that a dog’s placental barrier is a more efficient defense against cotinine compared to the human one. Further investigations will be required to confirm this hypothesis, also considering that cotinine concentration in serum and hair samples of both exposed bitches and newborn puppies were higher than values reported in pregnant women and babies [31,39]. It is noteworthy that pregnant women displayed an acceleration in nicotine and cotinine clearance (up to 140%), which is more pronounced at 18–22 and 32–36 weeks of gestation [40,41]. To date, the metabolism of nicotine and cotinine in canine species during pregnancy is unknown. Therefore, a species-specific nicotine metabolism may explain these differences between dogs and humans.

Pregnancy is associated with many physiologic changes that can influence drug absorption, distribution, and excretion, as happens for cotinine metabolism [41,42]. Hemodynamic changes accompanying the expansion of plasma volume increase cardiac output and glomerular filtration rate and enhance renal excretion of drugs [43]. For these reasons, we compared cotinine concentrations in pregnant and non-pregnant bitches. Albeit without a statistical significance, we recorded higher serum and hair cotinine concentrations in pregnant than non-pregnant bitches exposed to passive tobacco smoke, suggesting a different susceptibility to cotinine due to pregnant conditions. This aspect deserves more insight and should not be overlooked, since a possible greater receptivity to passive smoke in pregnancy could increase the risks for the unborn puppy.

In our study, all matrices were significantly correlated with each other, as reported in humans [31]. In particular, the maternal serum cotinine concentration was positively correlated with maternal hair, neonatal hair, and amniotic fluid. This aspect makes hair and amniotic fluid collected at birth handy alternatives to blood with a less invasive impact on sampling. As expected, based on the literature in humans and dogs [20,25,30], bitches with smoking owners had about six-fold greater blood cotinine concentrations than not exposed bitches. Similarly, maternal hair cotinine was 7.2 times higher in exposed compared to not exposed dogs.

With regard to the clinical effects of smoke exposure in pregnant bitches, no correlation was found with respect to the parameters considered, that is, maternal age, body weight, litter size, neonatal mortality, and malformations. The literature is conflicting about the impact of age on cotinine concentration in adult smokers [44], while young children seem to have higher levels than older/adolescents, possibly due to differences in cotinine metabolism and clearance [45]. In agreement with studies in humans and rats [46,47,48], we did not find correlations between cotinine concentration in any matrices and body weight and litter size. On the contrary, an increased risk of perinatal mortality (stillbirth and early neonatal mortality) has been reported in smoking and passive-smoking pregnant women [49], but not in the dogs of our caseload. In addition, a racial difference in cotinine levels among children exposed to environmental tobacco smoke has been argued. Some authors hypothesize that the racial preference for some brands of cigarettes with different additives could explain this aspect which, however, is not applicable to the canine species [50]. The small sample size and heterogeneity of this caseload limited the generalization of these results. Additional studies are desirable to verify the consequences on birth weight and other neonatal outcomes. For example, in humans, a link between maternal smoking and cleft lip has been suggested and nicotine is a suspected fetal neuroteratogen [12,41,51]. Maternal smoking during pregnancy may increase, by nearly seven-fold, the risk of cleft lip and/or palate in babies due to the lack of enzymes involved in the biotransformation of toxic compounds derived from tobacco [52]. It may be too early to talk of the effects of passive smoking on the health of newborn puppies, however, comorbidity in malformations and neonatal mortality in brachycephalic breeds cannot be excluded because their short nose reduces their capacity for cigarette smoke filtering [28]. Similarly, tobacco smoke exposure could also provide new interpretations to the increasing forms of allergy, intolerance, or immunosuppression diagnosed in dogs. Future studies should also be targeted at verifying any association with the anogenital distance (AGD), the distance from the center of the anus to the genitals, in puppies exposed to tobacco smoke during pregnancy. In fact, AGD is a sensitive biomarker of prenatal androgen and antiandrogen exposure in female infants of smoking mothers besides being involved in hypospadias and cryptorchidism [53,54]. In humans, prenatal exposure to endocrine disruption factors such as tobacco smoke during the masculinization period of development is associated with shorter AGD in male children, while a significant increase in AGD is reported in female infants [53,54]. Even in dogs, the administration of synthetic androgens during pregnancy had significant implications on genital development. In particular, female offspring showed an increased anogenital distance along with bilateral ovotestis and a penis-like structure. However, no study deepened the role of smoke exposure on gonadal masculinization in the canine species so far [55].

## 5. Conclusions

Amniotic fluid, blood, and hair are reliable biospecimens for assessing canine exposure to smoke. Apart from easy storage (r/t) and sample analysis, hair would be preferred for measuring cotinine concentration due to the simplicity of collection and its non-invasiveness, mainly in newborn puppies. Preliminary, new, and interesting cues emerged from this study on cotinine detection in dams and puppies exposed to tobacco smoke during pregnancy. Cigarette smoking is the foremost modifiable risk factor for adverse pregnancy outcomes. Pediatric patients are especially vulnerable to second-hand smoke due to narrower bronchi, increased respiratory rate, and immature immune systems. A greater awareness of dog breeders and owners on the risks related to smoke exposure in pets is recommendable, mainly during pregnancy and the growth phase of the puppy. In fact, nicotine and some other chemicals in cigarette smoke could pass even into the milk, reducing its production and quality and affecting the health of the suckling baby [56].

## Figures and Tables

**Figure 1 vetsci-10-00321-f001:**
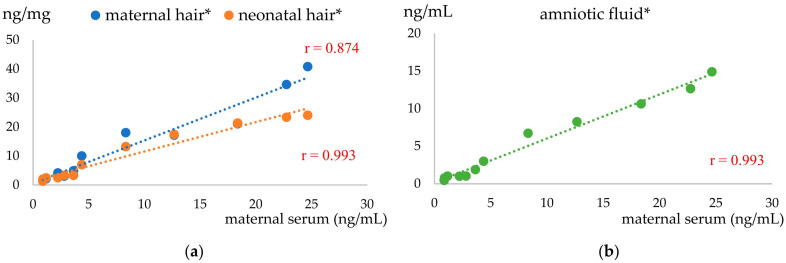
Correlation between cotinine concentration in maternal serum and other substrates: (**a**) Blue line and dots indicate cotinine concentration in maternal hair and orange line and dots indicate cotinine concentration in neonatal hair; (**b**) Green line and dots indicate cotinine concentration in amniotic fluid; * means *p* < 0.0001.

**Figure 2 vetsci-10-00321-f002:**
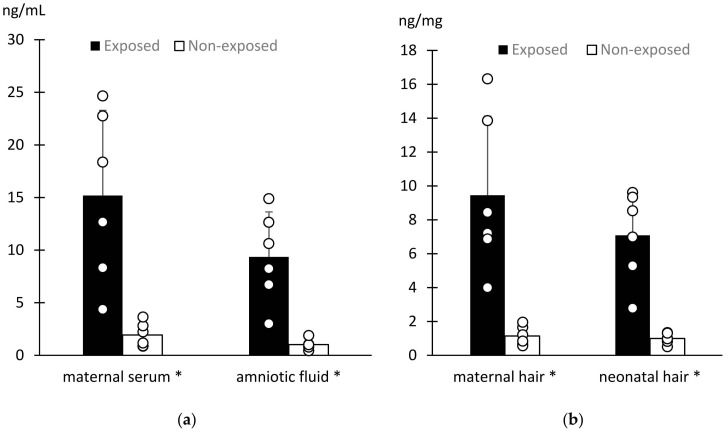
Cotinine concentrations in exposed and not exposed pregnant bitches and puppies. Bars show the mean ± sd while scatter plots show the single value of cotinine concentrations in maternal serum and amniotic fluid (**a**) and in maternal and neonatal hair (**b**). * means *p* = 0.004.

**Figure 3 vetsci-10-00321-f003:**
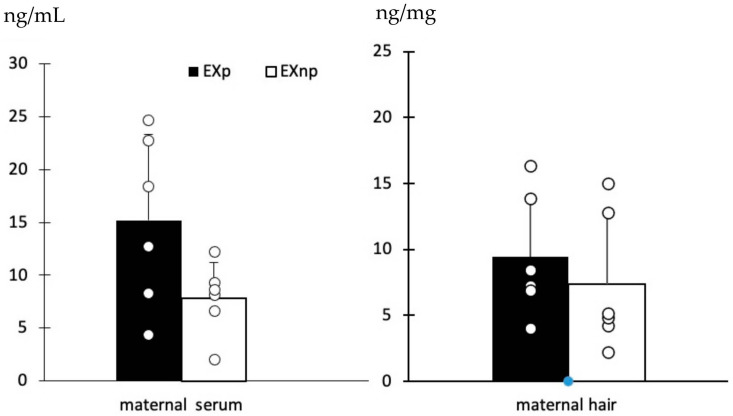
Cotinine concentrations in pregnant and non-pregnant bitches exposed to tobacco smoke. Bars show the mean ± sd while scatter plots show the single value of cotinine concentrations in maternal serum and hair. EXp means pregnant bitches exposed to cigarette smoke; EXnp means non-pregnant bitches exposed to cigarette smoke.

**Table 1 vetsci-10-00321-t001:** Dogs’ population.

Breed	Age (ys)	BW (kg)	Smoke Exposure	Intensity	Pregnant	Litter Size
Ambully	2	28.3	NEp		yes	5
Bouledogue	2	17.65	NEp		yes	8
Bernese Mountain D.	4.5	63.6	NEp		yes	3
German Shepherd	8	30.5	NEp		yes	4
Kurzhaar	5	33	NEp		yes	2
Staffordshire Bull T.	2	14	NEp		yes	6
Bernese Mountain D.	2	45.9	EXp	+	yes	8
Basset-hound	2	31.5	EXp	++	yes	8
Basset-hound	4.5	32.3	EXp	++	yes	3
Entlebucher Mountain D.	4.5	27.5	EXp	+	yes	3
Bernese Mountain D.	3.5	48	EXp	+	yes	8
Bernese Mountain D.	3	54.7	EXp	+	yes	3
Basset-hound	6	23	EXnp	++	no	
Basset-hound	3.5	24	EXnp	++	no	
Basset-hound	1.5	21	EXnp	++	no	
Basset-hound	2	28	EXnp	+	no	
Mongrel	12	15	EXnp	++	no	
Mongrel	11	30	EXnp	++	no	

BW means body weight; ‘+’ means up to 5 cigarettes per day; *“*++*”* means more than 5 cigarettes per day; NEp means pregnant bitches not exposed to passive tobacco smoke; EXp means pregnant bitches exposed to passive tobacco smoke; EXnp means non-pregnant bitches exposed to passive tobacco smoke.

## Data Availability

All the data that support the findings of this study are available from the corresponding author.

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
