# Peer review of "Tobacco Smoke Exposure in Pregnant Dogs: Maternal and Newborn Cotinine Levels: A Pilot Study"

_vetsci, 2023, doi:10.3390/vetsci10050321_

Round 1

Reviewer 1 Report

This study has investigated cotinine levels in dogs from smoking and non-smoking environments. The matrices examined are maternal serum, maternal hair, neonatal hair and amniotic fluid. While sample sizes are low, the study reveals that cotinine is detectable in all matrices and that there is a correlation between maternal serum and hair concentrations (maternal, neonatal), and also a correlation between maternal serum cotinine and that found in amniotic fluid. Hair concentrations were also measured in non-pregnant exposed dogs. The study is an interesting one and my questions are of a minor nature.

1. The correlation data suggests a linkage between maternal and amniotic fluid levels (also hair). Can the authors provide reassurance that litter size and breed were not confounders. 

2. Were the highest tissue cotinine levels found in households where more cigarettes had been smoked (> vs < 5 cigarettes per day).

3. Please provide the r values for the correlation studies.

4. Please present figures 2 and 3 as scatter plots where we can see each individual value. Also, present means and SEMs. his gives a much better impression of the spread of data.  

5. Did the authors measure anogenital distance (AGD) in the pups and were there any differences between male and female pups, particularly in terms of amniotic fluid concentrations? Linkages between smoking exposure and AGD have been reported in the human.

6. Regarding differences between data presented here and in the human literature, the theory around the different placentas is interesting. Would the authors anticipate any metabolic differences between humans and dogs? Refer to papers from the group of Fowler P and colleagues.

Author Response

This study has investigated cotinine levels in dogs from smoking and non-smoking environments. The matrices examined are maternal serum, maternal hair, neonatal hair and amniotic fluid. While sample sizes are low, the study reveals that cotinine is detectable in all matrices and that there is a correlation between maternal serum and hair concentrations (maternal, neonatal), and also a correlation between maternal serum cotinine and that found in amniotic fluid. Hair concentrations were also measured in non-pregnant exposed dogs. The study is an interesting one and my questions are of a minor nature.

We thank the reviewer for the valuable comments and suggestions that help us to improving the manuscript.

  1. The correlation data suggests a linkage between maternal and amniotic fluid levels (also hair). Can the authors provide reassurance that litter size and breed were not confounders.

R/ Thanks for point this aspect which deserves further investigation. In agreement with previous studies on rat model reporting no impact of nicotine dose in litter size (von Chamier et al. 2021; Breit et al., 2022), we didn’t found  correlation between cotinine concentration in any matrices and litter size. However, the small sample size and heterogeneity of breeds of our caseload prevent us from further generalization.

  1. Were the highest tissue cotinine levels found in households where more cigarettes had been smoked (> vs < 5 cigarettes per day).

R/ We found no correlation between the number of cigarettes and cotinine concentration. However, for future studies the daily exposure time will be taken into account as well as the number of cigarettes as it could provide interesting insights.

  1. Please provide the r values for the correlation studies.

R/ We added the r values in Figure 1.

  1. Please present figures 2 and 3 as scatter plots where we can see each individual value. Also, present means and SEMs. his gives a much better impression of the spread of data.  

R/ Done. Mean ± sd has been inserted in Figure 2 and 3, and the results have been presented also as scatter plots in Figure 4 and 5.

  1. Did the authors measure anogenital distance (AGD) in the pups and were there any differences between male and female pups, particuarly in terms of amniotic fluid concentrations? Linkages between smoking exposure and AGD have been reported in the human.

R/ Thanks for underline this aspect. It is a very interesting point that we hadn’t addressed. A possible link between AGD and smoking exposure as well as cryptorchidism - as reported in humans- has now been added to the discussion.

  1. Regarding differences between data presented here and in the human literature, the theory around the different placentas is interesting. Would the authors anticipate any metabolic differences between humans and dogs? Refer to papers from the group of Fowler P and colleagues.

R/ Thanks for your valuable suggestion. A brief mention of the different metabolic hypotheses related to the type of placentation was included in the discussion.

Reviewer 2 Report

Reviewer(s)' Comments to Author:
Manuscript ID: Vetsci-2331641

Title: Tobacco smoke exposure in pregnant dogs: maternal and newborn cotinine levels. A pilot study

Authors: Giulia Pizzi , Silvia Michela Mazzola , Alessandro Pecile , Valerio Bronzo , and Debora Groppetti

The manuscript provides a significant contribution to our knowledge about tobacco smoke exposure in dogs. The main aim was to reveal the harmful effects of smoke on newborn dogs by determining the cotinine level in the serum and hair of dogs living with smoking or non-smoking owners.

The aims, sampling procedure, and methods of data analysis are clearly stated and introduced. The statistical analyses are appropriate. The results and facts are presented clearly and sufficiently fully and are separated from interpretations. The authors know well the literature on the subject and fairly discuss the correspondence of results.

Given the non-invasive nature of hair testing, I recommend for the next study include 6 more dogs that are not pregnant and live with a non-smoking owner. Thus, they would form an absolute control group.

Further remarks:

I recommend entering new keywords because the current ones can all be found in the title: „dog, cotinine, newborn”. This way the new, changed keyword will help others find the article.

Maybe nicotine could also be included as a keyword, although this substance was not the subject of the study.

Figure 2: The presentation of the diagrams could be clearer, the column with white dots is particularly difficult to see in print.

Author Response

The manuscript provides a significant contribution to our knowledge about tobacco smoke exposure in dogs. The main aim was to reveal the harmful effects of smoke on newborn dogs by determining the cotinine level in the serum and hair of dogs living with smoking or non-smoking owners.

The aims, sampling procedure, and methods of data analysis are clearly stated and introduced. The statistical analyses are appropriate. The results and facts are presented clearly and sufficiently fully and are separated from interpretations. The authors know well the literature on the subject and fairly discuss the correspondence of results. 

Given the non-invasive nature of hair testing, I recommend for the next study include 6 more dogs that are not pregnant and live with a non-smoking owner. Thus, they would form an absolute control group.

R/ The authors thank the Reviewer for the positive comments and suggestions.

Further remarks:

I recommend entering new keywords because the current ones can all be found in the title: „dog, cotinine, newborn”. This way the new, changed keyword will help others find the article.

Maybe nicotine could also be included as a keyword, although this substance was not the subject of the study.

R/ Done.

Figure 2: The presentation of the diagrams could be clearer, the column with white dots is particularly difficult to see in print.

R/ The colors of the diagrams have been changed to be more clear.

Reviewer 3 Report

The evaluation of the effects of cigarette smoke in pregnant dogs is a topic of interest to many readers. It's very interesting and I have a few questions and recommendation. 

1) Dogs indirectly exposed to cigarette smoke are not only affected by the number of cigarettes they were exposed to per day, but perhaps also the duration of exposure. Have you checked whether dogs was continuously exposed to cigarette smoke before mating, even before conception?

2) Have you compared statistically whether there is any significance in whether or not you are pregnant according to the amount of exposure per day?

3) Even if exposed to cigarette smoke, a pregnant object is confirmed in a mother with 5 or less per day. It is thought that significant results can be obtained by comparing the average number of litters of the breed of individuals exposed to tobacco smoke and those not exposed.

Author Response

The evaluation of the effects of cigarette smoke in pregnant dogs is a topic of interest to many readers. It's very interesting and I have a few questions and recommendation. 

We thank the Reviewer for the comments

1) Dogs indirectly exposed to cigarette smoke are not only affected by the number of cigarettes they were exposed to per day, but perhaps also the duration of exposure. Have you checked whether dogs was continuously exposed to cigarette smoke before mating, even before conception?

R/ We agree. The inclusion criteria was an indoor consumption by the owner of at least one cigarette per day in the last two months. Therefore, all dogs in our study have been exposed at least from conceptus. Although the number of cigarettes was not related, we cannot exclude that the daily exposure time could affect cotinine concentration. However, we cannot speculate on this as we haven not considered it.

2) Have you compared statistically whether there is any significance in whether or not you are pregnant according to the amount of exposure per day?

R/ Sorry, not sure I understand your question. We have not explored the impact of smoke exposure on pregnancy rate, for which much more dogs would have been needed. This is a very interesting point that deserves further investigation.

3) Even if exposed to cigarette smoke, a pregnant object is confirmed in a mother with 5 or less per day. It is thought that significant results can be obtained by comparing the average number of litters of the breed of individuals exposed to tobacco smoke and those not exposed.

R/ Sorry again, not sure I understand your question. Is your suggestion to compare litter size between exposed and non-exposed dams based on the average number of puppies expected for the breed? This aspect is certainly very interesting and could show a negative effect of passive smoking on prolificacy. However, our caseload, again, is too small to permit a conclusion on this point. Moreover, we have only included dogs undergoing elective C-section which may be a bias for this comparison.